# Excellent Oxygen Barrier Property of Unfilled Natural Rubber/*trans*-Butadiene-co-Isoprene Rubber Vulcanizates under the Synergistic Effect of Crosslinking Density and Crystallization

**DOI:** 10.3390/polym16030345

**Published:** 2024-01-27

**Authors:** Pengcheng Xia, Huafeng Shao, Aihua He

**Affiliations:** Shandong Provincial Key Laboratory of Olefin Catalysis and Polymerization, Key Laboratory of Rubber-Plastics (Ministry of Education), Qingdao University of Science and Technology, Qingdao 266042, China

**Keywords:** oxygen barrier property, unfilled rubber, crosslinking density, morphology

## Abstract

The thermo-oxidative aging of rubber products is inevitable during their use and leads to product failure and can even endanger safety. Oxygen is an important factor that cannot be ignored during the thermo-oxidative aging process. Thus, the gas barrier property of rubber products is of significant concern. In this work, a strategy of crystallizing rubber in unfilled rubber composites was designed by firstly constructing a dual synergistic mechanism of crosslinking density and crystallization on the oxygen barrier properties. As a crystallizable polymer, *trans*-butadiene-co-isoprene rubber (TBIR) shows dendritic fibril crystals or spherulites in natural rubber (NR)/TBIR vulcanizates. Meanwhile, the vulcanizates containing TBIR have a higher crosslinking density than NR vulcanizates. These TBIR-rich crystals and high-crosslinking-density structures are distributed in vulcanizates like continuous islands. Contrary to what has been reported in the literature, the decrease in oxygen permeability of NR/TBIR is not only due to the high crosslinking density and free volume of the polymer matrix, but more importantly, the spherulites of TBIR play a role in blocking and prolonging the oxygen diffusion path during the diffusion of oxygen in the polymer composites. We propose that the compatible crystalline polymer can replace the lamellar filler, play the role of the oxygen barrier in rubber composites, reduce the diffusion and dissolution of oxygen, and achieve the effect of improving the thermo-oxidative aging property of the rubber composite. Future research will follow the morphology evolution of TBIR crystals, their crosslinking structure and density, and interactions between TBIR and NR on the oxygen barrier and thermo-oxidative aging property.

## 1. Introduction

As the only part that is in direct contact with the ground, the safety of tires has become the most important factor affecting the driving stability of vehicles. To ensure the comfort and safety of the vehicles while driving, maintaining the airtightness of pneumatic tires is crucial [1]. During the service process, especially under the effects of heat, O_2_, pressure, and the natural environment, rubber products are vulnerable to aging and degradation through the attack of oxygen, resulting in reduced service life and even safety [2]. Adding antioxidants is undoubtedly the most direct way to improve the thermo-oxidative aging performance of rubber products [3]. However, traditional antioxidants are mostly organic molecules which are volatile substances, have limited solubility, and can easily migrate from the rubber matrix to the surface, thereby reducing the thermo-oxidative resistance and causing environmental pollution. Increasing the average molar mass, or grafting to macromolecules, could reduce the migration of antioxidants [4,5]. Many researchers have reported on the synergistic antioxidant system of Zinc methacrylate (ZDMA) and traditional antioxidants. Under the action of ZDMA, the small molecule antioxidants and rubber double bonds form copolymers through the addition reaction, solving the migration of antioxidants and improving the thermo-oxidative aging properties of rubbers such as ethylene-propylene-diene monomer (EPDM) and nitrile butadiene rubber (NBR) [1,2].

As has been reported, the thermal-oxidative aging process consists of two inseparable parts: oxygen penetrating the rubber and oxygen reacting with the rubber [6]. The lower the oxygen penetration, the smaller the amount of oxygen entering the rubber. Furthermore, the subsequent reaction of oxygen with the rubber is also reduced. Lamellar fillers, due to their lamellar structure, large specific surface area, and gas barrier properties [7], can be used to improve the thermo-oxidative aging properties of rubber products [3]. Compared with spherical fillers such as carbon black, Lamellar fillers, like clay [7,8], montmorillonite (MMT) [3,9], and graphene [10,11,12,13,14] play a role in oxygen isolation. Thus, low oxygen permeability and the capture of free radicals are conducive to improving the thermo-oxidative aging resistance of diolefin rubber. Combining fillers with traditional antioxidants to improve the migration of antioxidants has also become a focus of research into improving the thermo-oxidative aging performance of rubber. Antioxidant-functionalized modified graphene [15], carbon nanotubes [16], and ortho aniline (antioxidant RT)-intercalated zirconium phosphate [17] have been used to improve the thermo-oxidative aging performance of polyolefins and styrene butadiene rubber (SBR). The introduction of rectorite, or organically modified MMT, into NR increases the diffusion path of nitrogen, resulting in lower gas permeability than pure NR. Furthermore, the selective permeability of gas can be realized by adjusting the filler type, solubility coefficient, diffusion coefficient, and permeability coefficient of different gases such as nitrogen, oxygen, and carbon dioxide [18,19]. However, the aggregation and stress concentration of filler in the rubber matrix has always been a problem due to the different polarities and high surface energy [3,17,20].

High-barrier rubber used in rubber tires can maintain the tire pressure at a constant value, greatly improving the service life of tires, controllability of transportation, and safety stability, achieving economic, green, and safe travel for humans. At present, high-speed tires are tubeless designs, and the innermost layer of the tire is currently made of butyl rubber (IIR) or brominated butyl rubber (BIIR), which have the best air tightness [13,21,22]. Recently, the improvement of the gas barriers of non-butyl rubber, especially the oxygen barrier, has aroused more and more interest. Blending is the most commonly used method to improve the properties of polymer. This can inspire us to blend lower-air-tightness rubber with IIR or BIIR [23,24]. Rattanasom reported that the elongated particles of the BIIR in the dispersed phase in filled NR/BIIR can serve as a gas barrier and reduce the gas permeability significantly [25]. For vulcanized rubber, the crosslinking can also improve the gas barrier of the rubber [26]. The increase in crosslinking density results in a decrease in free volume, prolonging the diffusion path of the gas and, thus, improving the barrier properties of the matrix [27,28,29].

To the best of our knowledge, the preparation of the unfilled rubber with high gas barrier properties through adjusting the crystalline morphology and crosslinking structure of the rubber has not been reported. In our previous work, we have reported that TBIR can serve as a compatibilizer to improve the compatibility of NR/BR systems and enhance the fatigue resistance and thermo-oxidative aging resistance of the rubber matrix [30,31]. In this study, for the first time, unfilled NR/TBIR vulcanizates with low oxygen permeability were shown to improve the oxygen barrier properties of unfilled rubbers via the dual synergistic mechanism of crystallization and crosslinking density. TBIR crystalline phases and crosslinked TBIR-rich phases as aggregated spherulites, and higher crosslinking density domains as fillers, could obstruct oxygen diffusion and dramatically decrease the oxygen permeability of the rubber matrix.

## 2. Experimental

### 2.1. Materials

NR was procured from Jinghong Manpu Rubber Co., Ltd., Wujiang, China (SCRWF, Mooney viscosity is 80.2@100 °C). TBIR was purchased from Shandong Huaju Polymer Materials Co., Ltd., Binzhou, China (TBIR0713, Mooney viscosity was 50.4@100 °C, F_Bd_ = 21.3%, *trans*-1,4-unit content of Ip and Bd units was above 94%). Zinc oxide, antioxidants (N-1,3-dimethylbutyl-N’-phenyl-p-phenylenediamine (antioxidant 4020), poly(1,2-dihydro-2,2,4-trimethyl-quinoline) (antioxidant RD)), accelerator (N-cyclohexylbenzothiazole-2-sulphenamide (CBS-80)), sulfur (S-80), and toluene were commercial grade and used as received.

### 2.2. Samples Preparation

NR and TBIR were rolled 8–10 times in the two-roll open mill at 50 °C and 70 °C individually before mixing. Two-stage mixing in the internal mixer was chosen to ensure sufficient mixing of the rubber material and additives. The formulations were NR/TBIR = 0/100, 80/20, 40/60, 80/20, and 0/100. The first stage of mixing was performed at 70 °C and 70 rpm for two minutes to mix the NR and TBIR. Then, 4.4 phr of zinc oxide, 1.5 phr of antioxidant 4020, and 1.2 phr of antioxidant RD were added and mixed for another 8 min. After the first stage of mixing, the rubber compounds were taken out of the internal mixer below 150 °C and stored at room temperature for 30 min. In the second stage, the as-obtained rubber compounds were added into the internal mixer at 60 °C and 30 rpm for 2 min, and then 1.88 phr of sulfur and 2.0 phr of accelerator CBS were added and mixed for another 4 min below 90 °C. Finally, the rubber composites in sheets were obtained using the two-roll open mill at 50 °C. After being stored at room temperature for different times, the as-obtained rubber composites were cured at 150 °C for optimum curing time (t_90_+3 min) under 10 MPa pressure.

### 2.3. Characterization and Measurements

The vulcanization characteristic curve (MDR Premier Rotorless Vulcanizer, Alpha Corporation, Dulles, VA, USA) of the samples was measured at 150 °C and 50 min under a pressure of 10 MPa to determine the vulcanization time.

The apparent crosslinking density of the vulcanized rubber sample was tested using the equilibrium swelling method. A total of 20–25 mg of the sample was placed in 20 mL of toluene at 30 °C for 5 h until swelling equilibrium was achieved. The crosslinking density (*Vr*) was calculated according to the formula
(1)$$V_{r}=\frac{1}{1+(\frac{m_{a}}{m_{b}}-1)\times \rho_{r}/(\alpha \times \rho_{s})}$$
where *m_a_*-sample mass before swelling, mg; *m_b_*-sample mass after swelling, mg; *ρ_r_*-rubber density, mg/cm^3^; *ρ_s_*-solvent density, mg/cm^3^; *α*-mass fraction of rubber in the composite.

A Fourier Transform Infrared Spectrometer (FTIR, TENSOR II, Bruker, Germany) was used for confirming the structure of samples. DMA was conducted on a DMA Q800 dynamic mechanical thermal analyzer (TA company, Boston, MA, USA) with a tensile mode.

Polarized optical microscopy (POM) images were observed using an Olympus (BX 51-P, Tokyo, Japan) optical microscope equipped with an Olympus camera (DP26). AFM was performed on a tapping mode with an Agilent Technologies 5500 atomic force microscope, Tempe, AZ, USA.

The O_2_ gas permeability was measured via the differential pressure method using an automatic gas permeability tester (VAC-V3, Jinan Languang, Jinan, China) at 23 °C and 39.3% RH in humidity. In detail, a sample sheet about 1 mm in thickness was fixed between two test chambers, upstream and downstream. The pressure on one face of the sample was kept at a constant value (0.4 MPa), while the other face was exposed to vacuum (vacuumed above 17 h for lower than 136 Pa initially). Then, the gas molecules diffused through the sample sheet due to the driving force of the pressure gradient. The O_2_ gas permeability (*P*) was calculated from the transmission rate of O_2_ gas determined using a pressure sensor.

## 3. Results and Discussion

### 3.1. Oxygen Barrier Property

Aging, especially thermal-oxidative aging, is an inevitable problem in the use of rubber products. During the thermal-oxidative aging process, oxygen plays a very important role. As has been reported, the thermal-oxidative aging process consists of two inseparable parts: oxygen penetrating into the rubber and oxygen reacting with the rubber [6]. The lower the oxygen penetration, the smaller the amount of oxygen entering the rubber. Furthermore, the subsequent reaction of oxygen with the rubber is also reduced. Thus, the oxygen permeability of NR and NR/TBIR has been measured and the results are shown in Figure 1. 

It can be seen from Figure 1 that the oxygen permeability coefficients (P) of pure NR and TBIR unfilled vulcanizates are both very high, reaching 2.5 × 10^−14^ cm^3^·cm/cm^2^·s·Pa^−1^ and 2.7 × 10^−14^ cm^3^·cm/cm^2^·s·Pa^−1^, respectively. The oxygen permeability coefficient of the NR/TBIR unfilled vulcanizates is lower than that of the two pure vulcanizates. The oxygen permeability coefficient for NR/TBIR (80/20) is only 0.7 × 10^−14^ cm^3^·cm/cm^2^·s·Pa^−1^, which is 72% lower than that of the pure NR vulcanizate. It is worth noting that with the increase in TBIR, the oxygen permeability coefficient of the NR/TBIR vulcanizate slightly increased to 2.2 × 10^−14^ cm^3^·cm/cm^2^·s·Pa^−1^. The oxygen permeability coefficient of the NR/TBIR vulcanizate with 20 phr TBIR content is the lowest value among these samples. The permeability coefficient is strongly affected by the diffusion coefficient (D) and solubility coefficient (S), and the corresponding value can be multiplied directly by the D and S. Generally, the diffusion coefficient and solubility coefficient represent the movement and interaction of the small molecules (O_2_ in this work) in the polymer composites, respectively [32]. It is reported that there are several factors affecting the diffusion and solubility of the gas in polymers, such as the fraction free volume, cavitation, intermolecular interaction force, the mobility of the molecular chain, and so on [32]. 

Cohesive energy density (CED), which is defined as the energy required to completely remove a unit volume of molecule from its neighbors to infinity, can be used to quantify the molecular interaction. The CED of NR, TPI, and TPB is about 289 MPa, 275 MPa, and 213 MPa, respectively [33]. The CED of NR is larger than TBIR due to the higher polarity brought about by polar substances such as protein in the NR [32]. TBIR, as reported, is a multiblock copolymer containing a block trans-structure of butadiene and isoprene [30]. Thus, we inferred that the CED of TBIR is lower than NR. According to the references, the larger CED can result in higher interaction between the polymers and small molecules, which can lead to increased solubility of oxygen in the rubber matrix [32]. Consequently, the S of NR should be larger than that of TBIR. However, as can be seen from Figure 1, the S of NR is smaller than that of TBIR, where the D is reversed. And, interestingly, the D and S also decrease first, reach the minimum value when TBIR is 20 phr, and then increase correspondingly with the increase in TBIR in the NR/TBIR vulcanizates. Therefore, we speculate that it is not just the effect of CED on the D and S. To illustrate the reason why, the glass transition temperature (Tg) was tested.

### 3.2. DMA

As reported, the diffusion and dissolution of small molecules is closely related to the free volume in the polymer matrix [34]. Generally, S depends on free volume and molecular interaction between small molecules and polymers under the same temperature and pressure [35]. *T*g can also be used to characterize the mobility of chain segments. The lower *T*g usually implies higher chain mobility and bigger free volume [36]. As discussed above, the CED of TBIR is lower than NR. The lower the CED, the lower the *Tg*. As shown in Figure 2, the NR/TBIR and TBIR vulcanizates show decreased *Tg* compared with the NR vulcanizate. This means that the ability of movement of the chain and free volume in vulcanizates containing TBIR is greater than that of NR. The sum of the increased TBIR content in NR/TBIR would reduce the *Tg* and increase the free volume, which is beneficial as it increases the diffusion performance of oxygen in the polymer matrix [32]. However, the oxygen permeability of NR/TBIR is lower than that of NR (Figure 1). Therefore, we speculate that the factors affecting permeability are not just free volume, *Tg*, and CED. This may potentially be linked to the crystallinity of TBIR in the vulcanizates [37]. As reported, the TBIR crystals act as the fillers to block the movement of oxygen [38], which results in the intensifying of intermolecular interaction between the oxygen and polymer matrix.

### 3.3. Curing Characteristics and Crosslinking Density

The curing torque curves and crosslinking density are shown in Figure 3 and listed in Table 1. Zhang reported that the heterogeneously crosslinked rubbers in unfilled rubbers exhibited a sea–island morphology, in which the dispersed, densely crosslinked rubber granules showed low N2 permeability and inhibited the gas permeation in the rubber matrix [28]. The minimum torque value (ML) of NR is 1.40 dNm. The increased ML can be observed with increasing TBIR in NR/TBIR due to the high viscosity and rigidness of TBIR. Increased t_10_ and t_90_ indicate an improvement in processing safety and a slower vulcanization rate with increasing TBIR addition. The torque difference (MH-ML), which reflects the crosslinking density indirectly [31], exhibits a slight decrease with the TBIR fraction. The change in crosslinking density is in keeping with the torque difference. As shown in Figure 3b, the crosslinking density of the pure NR and TBIR is 0.1785 × 10^−3^ and 0.2115 × 10^−3^ mol/cm^3^, respectively. The crosslinking density of the NR/TBIR vulcanizates increases to about 0.2163~0.2179 × 10^−3^ mol/cm^3^ with the increase in TBIR content. This phenomenon can be attributed to the fact that TBIR can enhance the uniform dispersion of processing agents between the NR and TBIR, leading to the increase in crosslinking density [30]. Previous studies [30,31] have confirmed that TBIR is prone to forming co-vulcanization bonds with NR during the vulcanization process, making it easier to form a three-dimensional crosslinking network structure compared to homogeneous polymers.

### 3.4. Crystallinity of TBIR

TBIR, a multi-block copolymer with different length *trans*-isoprene blocks (TPI) and *trans*-butadiene blocks (TPB) in the same molecular chain, is a crystalline rubber and can be either in an unvulcanized state or vulcanized [39,40,41]. According to our previous work [30,31], TBIR can improves the thermo-oxidative aging property of unfilled/filled NR/BR vulcanizates through the synergistic effects of TBIR fibrous crystals and uniform filler dispersion. Figure 4 shows an FTIR spectra range from 600 cm^−1^ to 1600 cm^−1^ for NR, TBIR, and NR/TBIR. An overlapping shoulder absorption peak at 1030 cm^−1^ can be attributed to the α-form crystal of TPI blocks [42]. And, the absorption peak at 1450 cm^−1^, attributed to the asymmetric deformation vibration of methyl, is not affected by crystallization or vulcanization and can be used as the internal peak [43]. Consequently, the peak area ratios of A_1030_/A_1450_ can be marked to quantitatively evaluate the relative crystalline content of α-form crystals.

From Figure 4 it can be observed that NR/TBIR (80/20) has the highest relative crystalline content. As the TBIR fraction increases, the relative crystalline content decreases, but the difference in relative crystalline content between TBIR and NR/TBIR with high TBIR content (40/60 and 20/80) is not significant. From this phenomenon, it can be deduced that during the vulcanization process, pre-vulcanized NR may play the role of nucleating agent to promote and improve the crystallization of TBIR.

Figure 5 shows the effect of TBIR content of NR/TBIR on oxygen permeability, crystallinity, and crosslinking density. The crosslinking density of the NR vulcanizate was lower than that of the TBIR and NR/TBIR vulcanizates, and no crystallization was observed in the NR vulcanizate (DSC and XRD results not provided). The oxygen permeability coefficient of the NR vulcanizate was higher than the NR/TBIR vulcanizates (about 2.5 × 10^−13^ cm^3^·cm/cm^2^·s·Pa^−1^).

The crosslinking density remained basically unchanged with the increase in TBIR content in the NR/TBIR vulcanizates, but the crystallinity showed an obvious decreasing trend. As a result, the oxygen permeability of the NR/TBIR vulcanizates increased with the decrease in crystallinity. It can be deduced that TBIR crystals can play a similar role as fillers to inhibit the permeation and retard the diffusion of oxygen in the rubber matrix. As shown in Figure 1, the D and S of NR/TBIR reduced simultaneously when TBIR was added to the NR, resulting in reduced P. The result is improved oxygen barrier properties. The pure TBIR vulcanizate has the biggest oxygen permeability coefficient. The crystallinity of the TBIR vulcanizate is similar to NR/TBIR vulcanizates containing 80 phr TBIR, but the crosslinking density is lower than that of NR/TBIR. So, the NR/TBIR vulcanizates combined with TBIR improved the oxygen barrier properties due to the synergistic effect of crystallization and crosslinking density.

### 3.5. Morphology

As discussed above, the crystallinity of TBIR affected the oxygen permeability. It is well known that fillers, especially aligned lamellar fillers, improve the gas barrier property due to the prolonged path of gas diffusion [9,38,44,45]. Our previous work proved that the TBIR component or block does yield spherulites [46] or lamellar fibrous crystals [40] several nanometers thick and 500~1000 nm in length in the SSBR/BR/TBIR blend [37]. As shown in Figure 6, there is no obvious crystallization in the NR vulcanizate (Figure 6a,a’) and vulcanizated TBIR shows tiny crystals in the whole field of vision (Figure 6b’). The AFM image shows that there are dendritic fibril crystals in the TBIR. It is worth noting that the NR/TBIR (80/20) blend exhibits relatively obvious spherulites with typical black cross patterns. The size of the spherulites is about 10–20 μm (Figure 6c’). The AFM images also show the big spherulites but no fibrous crystals in NR/TBIR with 20 phr TBIR. When the content of TBIR is further increased, the spherulites become fibrous crystals, like the TBIR vulcanizate (Figure 6). As shown in Figure 4, TBIR has the biggest crystallinity in NR/TBIR (80/20) among these samples. Thus, we hypothesize that NR may have acted as a nucleating agent to accelerate and complete the TBIR spherulites, resulting in increased crystallinity and, subsequently, reduced oxygen permeability. 

### 3.6. Evolution of Oxygen Barrier

Figure 7 shows a comparison of the gas permeability of this work with IIR, NR, and their derivatives and filled or blending modifications from the references. The gas permeability is related to the gas LennardJones diameter. Oxygen has a smaller Lennard-Jones diameter of 0.347 nm than that of Nitrogen (0.38 nm) [34,38], resulting in the gas barrier of O_2_ being more difficult than N_2_ for the same polymer or polymer composite. It can be seen from Figure 5, for IIR, that the gas permeability of O_2_ is higher than that of N_2_ [20,24,47] in general. As the rubber with the bet gas barrier, the nitrogen permeability coefficient of IIR is about 1 × 10^−14^ cm^3^·cm/cm^2^·s·Pa^−1^. The nitrogen permeability coefficient of NR is about 8 × 10^−14^ cm^3^·cm/cm^2^·s·Pa^−1^ in our work, which is eight times the best of IIR among the references. What is unexpected is that the NR/TBIR vulcanizates show very excellent gas barrier properties of oxygen and nitrogen. The nitrogen permeability of NR/TBIR almost reached the level of IIR, about 2 × 10^−14^ cm^3^·cm/cm^2^·s·Pa^−1^. As is known, oxygen plays a vital role during thermo-oxidative aging. The oxygen permeability of NR/TBIR is about 0.7 × 10^−14^ cm^3^·cm/cm^2^·s·Pa^−1^*,* better than most of the reported rubber, such as IIR/PLLA(3 × 10^−14^ cm^3^·cm/cm^2^·s·Pa^−1^) [47], IIR(5 × 10^−12^ cm^3^·cm/cm^2^·s·Pa^−1^) [32], Re-IIR (30 × 10^−14^ cm^3^·cm/cm^2^·s·Pa^−1^) [48], IIR/MMT (10 × 10^−14^ cm^3^·cm/cm^2^·s·Pa^−1^) [9], cellulose reinforced NR (7.5 × 10^−13^ cm^3^·cm/cm^2^·s·Pa^−1^) [49], and NR/SBR (52 × 10^−12^ cm^3^·cm/cm^2^·s·Pa^−1^) [50], and almost reaches the level of some modified plastics such as EVA-OH [51] and PFS/SiO_2_ [34]. Thus, the obtained NR/TBIR vulcanizates are confirmed to have excellent gas barrier properties, including oxygen and nitrogen.

To the best of our knowledge, the preparation of unfilled rubber with high gas barrier properties by adjusting the crystalline morphology and crosslinking structure of the rubber has not been reported. Dual exploitation on both the designed and controlled crosslinking structure and aggregation structure of TBIR has been performed for the first time. As discussed above, the oxygen permeability of the NR/TBIR = 80/20 vulcanizate with 20 phr TBIR incorporation is more than three times lower (from 2.5 × 10^−14^ to 0.7 × 10^−14^ cm^3^·cm/cm^2^·s·Pa^−1^*)* than that of NR vulcanizate. To explain the significant increase in the oxygen barrier properties of the NR/TBIR vulcanizates, an illustration is shown in Figure 1. 

In the case of the NR/TBIR = 80/20 vulcanizate with TBIR incorporation, lots of TBIR spherulites with a size of less than 20 μm were generated and were prone to gather and cover the whole vulcanizates, while the crosslinking density of NR/TBIR vulcanizates was higher than the NR vulcanizate. The spherulites (pink ovals) and higher crosslinking density region (dashed circle) brought about by the TBIR in the NR/TBIR vulcanizates played a dual role in oxygen permeability, like the impermeable filler. The oxygen diffusion path is prolonged due to the obstruction of TBIR spherulites and higher crosslinking density. Compared with the NR vulcanizate with a relatively lower crosslinking density and noncrystalline, the oxygen permeability of NR/TBIR decreased.

When TBIR increased to 60 phr (NR/TBIR = 40/60 vulcanizate), it showed a small amount of large-sized fibrous crystals with large gaps between them (as shown in the AFM image). The solubility coefficient increased (Figure 1), resulting in an increase in oxygen permeating into the matrix. Meanwhile, due to the presence of such large-sized and loose fibrous crystals, the diffusion coefficient of O_2_ in the matrix was also increased, ultimately leading to an increase in the oxygen permeability coefficient. Therefore, we speculate that a large number of small-sized spherulites is more conducive to blocking the permeation and diffusion of oxygen, reducing the permeability coefficient and improving the barrier performance compared to loose fibrous crystals under a similar crosslinking density (Figure 3).

## 4. Conclusions

In summary, we have designed a novel dual strategy, stemming from the incorporation of crystalline TBIR into NR, to fabricate NR/TBIR composites with significantly decreased gas permeability and particularly superior oxygen barrier properties. Accompanied by TBIR crystalline phases as aggregated spherulites, the higher crosslinking density and TBIR-rich domains as fillers could obstruct oxygen diffusion, dramatically decreasing oxygen permeability. Future research should follow the morphology evolution of TBIR crystals, their crosslinking structure and density, and TBIR-NR/BR interactions on the oxygen barrier properties and the resulting thermo-oxidative aging property.

## Data Availability

Data are contained within the article.

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
