# Peer review of "Excellent Oxygen Barrier Property of Unfilled Natural Rubber/trans-Butadiene-co-Isoprene Rubber Vulcanizates under the Synergistic Effect of Crosslinking Density and Crystallization"

_polymers, 2024, doi:10.3390/polym16030345_

Round 1

Reviewer 1 Report

Comments and Suggestions for Authors

            Xia et al. describe methods to diminish rates of oxygen permeation and subsequently thermo-oxidation of tire rubber. This was performed by supplementing natural rubber (NR) with different ratios of trans-1,4-poly(butadiene-co-isoprene) (TBIR), manipulating their crystallizations and crosslinking densities, while maintaining the use of the typical antioxidants. This is an effort to create an improved rubber composite which can be filled with the traditionally used oxygen-isolating lamellar and/or spherical fillers to build long-lasting rubbers for tires or general industrial use. It was found that the NR may act as a nucleating agent for TBIR crystalline spherullites to form. The sample with the lowest oxygen permeability was the composite that had the highest relative crosslinking density and crystalline content, the 20 phr TBIR content rubber blend. 

General Writing Comments:

            In the Abstract and Introduction sections, the abbreviations NR, TBIR, EPDM, NBR, MMT, etc. should be defined the same way that IIR and BIIR were clearly defined. 

The Materials section does not mention purities, brands, or specifically from where the commercially available materials were gathered. 

The Samples preparation section fails to mention the ratios of NR and TBIR that were mixed. Also, the writing implies that the two stages of mixing take place in the internal mixer but then in lines 106 and 107 it is stated that the “as-obtained rubber blend was added into the internal mixer” as if it was not already in the internal mixer for the first mixing stage. 

The Results and Discussion Figure 1 analysis has some issues. On line 152, the measurement should read “2.5*10-14 cm3·cm/cm2·s·Pa-1.” On line 155, it should read “NR/TBIR (80/20).” “NR/TBIR (20/80)” does not make sense because the x-axis on the graph is a measure of TBIR to NR content and not NR to TBIR content; the measurement “7*10-14 cm3·cm/cm2·s·Pa-1” should read “0.7*10-14 cm3·cm/cm2·s·Pa-1” based on Figure 1. The measurement in line 158 should read ”2.2*10-14 cm3·cm/cm2·s·Pa-1” instead of “22*10-14 cm3·cm/cm2·s·Pa-1.” 

General Concepts:

Since the DMA analysis (Figure 2) does not agree with the data in Figure 1, it could be more useful to run a specific surface area (BET) test of the polymers to get a better idea of which materials have more free volume rather than finding the glass transition temperature alone. It seems like a more direct way (and more accurate way) of making these kinds of measurements. Since the material’s oxygen permeability was tested using the differential pressure method, there must be pores that can be measured with SSA to supplement the data that is already present. 

More trials should be performed with NR/TBIR ratios that change by smaller increments to get better oxygen permeability trend lines; it could be that the oxygen permeability gets even lower with a TBIR content of 10 or 30 phr. It could also be useful to perform some tests to see how water (liquid or vapor) may find its way into the material and harm its longevity. 

Since fillers are commonly used in rubbers to make them more durable and last a longer time, it could be effective to use fillers for the rubbers in the experiment rather than unfilled rubbers. This is so that they can be compared to types of rubbers that are already used commercially. There are no figures depicting the oxygen permeability of preexisting rubbers, so this could provide some degree of context to what is already being used to determine whether the materials made in this experiment are viable alternatives. 

Overall the manuscript is reasonable but still needs further work for publication.

Comments on the Quality of English Language

A bit difficult to read in a logical flow. Needs quite a bit of basic english writing throughout the manuscript. 

Reviewer 2 Report

Comments and Suggestions for Authors

The current work is interesting and enjoyed reading it. Below some minor comments/

L 42 better average molecular weight or even average molar mass

L 82 please also cite Nat. Mater. 2021 20, 1422

L 146 any literature data on oxygen concentrations ?

CED needs context for a general reader as one has theoretical values for crosslinked systems.

Ref 30 good to have extra refs if possible as key for the discussion.

Figure 6 please adapt size to fit one page and check labeling visibility

Figure 7 what are experimental errors?

Scheme 1 is nice but perhaps good for the introduction as well. Could the discussion be somewhat enlarged with more discussion of the spherulites (type and amount?)

Round 2

Reviewer 1 Report

Comments and Suggestions for Authors

Authors have effectively addressed reviewers concerns.